# Modification of Vegetation Structure and Composition to Reduce Wildfire Risk on a High Voltage Transmission Line

Tom Lewis [1,2,*], Stephen Martin [3] and Joel James [3]

1. Queensland Forest Consulting Services, Beerwah, QLD 4519, Australia
2. School of Environment and Science, Griffith University, Brisbane, QLD 4111, Australia
3. Powerlink Queensland, Virginia, QLD 4014, Australia; stephen.martin@powerlink.com.au (S.M.); joel.james@powerlink.com.au (J.J.)
* Correspondence: qldforestconsulting@proton.me

**Abstract**

The Mapleton Falls National Park transmission line corridor in Queensland, Australia, has received a number of vegetation management treatments over the last decade to maintain and protect the infrastructure and to ensure continuous electricity supply. Recent treatments have included 'mega-mulching' (mechanical mastication of vegetation to a mulch layer) in 2020 and targeted herbicide treatment of woody vegetation, with the aim of reducing vegetation height by encouraging a native herbaceous groundcover beneath the transmission lines. We measured vegetation structure (cover and height) and composition (species presence in 15 × 2 m plots), at 12 transects, 90 m in length on the transmission line corridor, to determine if management goals were being achieved and to determine how the vegetation and fire hazard (based on the overall fuel hazard assessment method) varied among the treated corridor, the forest edge environment, and the natural forest. The results showed that vegetation structure and composition in the treated zones had been modified to a state where herbaceous plant species were dominant; there was a significantly ($p < 0.05$) higher native grass cover and cover of herbs, sedges, and ferns in the treated zones, and a lower cover of trees and tall woody plants (>1 m in height) in these areas. For example, mean native grass cover and the cover of herbs and sedges in the treated areas was 10.2 and 2.8 times higher, respectively, than in the natural forest. The changes in the vegetation structure (particularly removal of tall woody vegetation) resulted in a lower overall fuel hazard in the treated zones, relative to the edge zones and natural forest. The overall fuel hazard was classified as 'high' in 83% of the transects in the treated areas, but it was classified as 'extreme' in 75% of the transects in the adjacent forest zone. Importantly, there were few introduced species recorded. The results suggest that fuel management has been successful in reducing wildfire risk in the transmission corridor. Temporal monitoring is recommended to determine the frequency of ongoing fuel management.

**Keywords:** overall fuel hazard; wildfire risk; forest edge; native species; plant composition

## 1. Introduction

Overhead transmission line networks are critical for ensuring safe and stable supply of electricity worldwide [1,2]. These lines often pass through forests and native vegetation in Australia, resulting in a network of linear corridors, where the vegetation is managed to ensure it does not reach the line infrastructure. In south-east Queensland, Australia, a 275 kV transmission line traverses Mapleton Falls National Park (MFNP), connecting

bulk generation from Central Queensland to Queensland's large population centres in the south-east. It is important that this infrastructure is protected to prevent broadscale outages that can impact on human lives and the provision of services (e.g., failure of life supporting equipment). Wildfire (also referred to as bushfire) represents a serious risk to transmission line infrastructure in Australia [3,4] and hence vegetation management of corridors is a high priority. The vegetation in MFNP varies but is predominantly open and tall open eucalypt forests (as defined by Specht, 1970 [5]). *Eucalyptus pilularis* tall open forests and *Eucalyptus racemosa* subsp. *racemosa*, *Lophostemon confertus*, *Syncarpia glomulifera*, *Eucalyptus acmenoides* open forests are common ecosystems adjacent to the transmission line corridor. These are highly flammable vegetation types [6]. Wildfires near electrical and transmission installations can lead to direct damage to the steel lattice towers, lines, and associated infrastructure [3]. Smoke and ash ionizing the air gap between the vegetation and the transmission lines, causing an arcing pathway ('flashovers'), can also damage infrastructure during a fire [7], and can endanger personnel fighting fires near transmission lines. This in turn can lead to major disruptions to electrical networks and the associated costs to human life and direct financial costs to industry [1,3]. Wildfires can also be ignited due to transmission line faults associated with vegetation contact during periods of high fire danger [8,9].

Fuel treatments beneath and nearby transmission lines is a priority to help ensure wildfires do not pose a safety risk to the community or interrupt the power supply. To maintain and protect the transmission line infrastructure, various fuel management methods are employed. In the MFNP transmission line corridor, a number of treatments have been applied over the last decade to prevent woody vegetation encroaching on the transmission lines and to reduce the risk of a wildfire damaging the infrastructure. An adaptive management trial, as a collaboration between Powerlink Queensland and the Queensland Parks and Wildlife Service, was initiated in 2018 to investigate the effectiveness of vegetation management treatments on fuel hazard and vegetation structure. Treatments have included a combination of mega-mulching (mechanical mastication of vegetation to a mulch layer), slashing, spot spraying, targeted tree felling, and planned burns. The most recent treatments comprised mega-mulching in August 2020, followed by a patchy, cool burn over part of the corridor and follow-up herbicide treatments (spot spraying) of the woody vegetation. The objective of these treatments was to modify vegetation composition and structure, to reduce fire risk while maintaining important ecological values, and, specifically, to alter the vegetation composition and structure in the corridor from vegetation dominated by taller native woody vegetation to lower-statured vegetation dominated by native grasses, sedges, herbs, and small shrubs.

Previous studies in transmission line corridors have shown that mechanical mastication or mega-mulching has been effective in modifying fuel structure and fuel connectivity for a period of up to 4 years [10,11]. Industry practitioners are encouraged to follow Integrated Vegetation Management approaches, rather than using mechanised mowing alone to manage large tracts of vegetation on transmission line easements [12]. Integrated management approaches, such as those applied in the MFNP corridor to date, may encourage certain biodiversity components. In fact, previous studies have highlighted the positive influence of corridors on plant species diversity at a landscape scale [13–17]. This is in accordance with the ecological literature that suggests that forest edges often have a higher diversity of plants, because edges can contain a mix of plant species from the forest interior and the adjacent vegetation type, partly due to the increased availability of resources, such as sunlight and wind, to aid seed dispersal and germination [18–21]. However, introduced weed species can also proliferate on transmission line clearings [22]. An international review by Franklin et al. (2021) reported that non-native species abundance and richness

was higher at the forest edge relative to the interior in 93% of studies where the edges were human-made [21]. In Australia, Beer (1999) reported that exotic ground layer species were more common on forest edges but declined within 7–16 m of the forest edge [23]. Given the location of the Mapleton corridor in a National Park setting, it is particularly important to ensure that management treatments do not encourage introduced weed species.

A previous project at the MFNP corridor by Sos et al. (2023) focused on using drone-derived imagery, photogrammetry, and geospatial analysis to determine the fuel structure beneath the transmission lines following mega-mulching treatments in 2020 [24]. This study recognised the potential for such remote sensing techniques to monitor the vegetation structure. Such techniques are particularly useful over large corridors, where access is challenging due to steep terrain. While such methods can monitor vegetation heights, the methods are yet to be refined to allow for remote determination of the vegetation species composition. As such, field surveys are needed to monitor the plant species growing in the transmission line corridor in this study. Relatively few studies have investigated the influence of human-created eucalypt forest edges on vegetation composition and structure. In fact, we are unaware of any published studies that have specifically investigated vegetation response to transmission line corridor management through native subtropical eucalypt forests.

Our study aimed to determine if fuel management activities have modified the vegetation structure and composition of a transmission line corridor, resulting in lowered wildfire risk. Specifically, we hypothesized that light-demanding grasses, herbs, and introduced weed species would be more frequently encountered in the managed areas and forest edges, relative to the adjacent forest.

## 2. Materials and Methods

This case study is located in Mapleton Falls National Park (MFNP), south-east Queensland, Australia (Figure 1), which is considered an ideal location because of the existing adaptive management trial. This National Park has a varied fire history, including planned burns and infrequent wildfires. Planned burning frequency has varied depending on the vegetation types and management objectives. The open eucalypt forests typical of the study area were burnt frequently as part of forest management activities (e.g., in some areas every three years) prior to the area becoming a National Park. Current management aims to conduct planned burns every 4–20 years, depending on the ecosystem and understorey species. The transmission line corridor through MFNP is approximately 70 m wide and runs approximately 3 km in length. There are a number of spans, which are the areas between each steel lattice tower along this corridor. The ground surface is undulating, with gradients varying along the corridor. To sample the variation in vegetation in a given span, a number of monitoring transects were established on different slope positions and with varying aspect. This study focused on a selection of 6 tower spans, which are considered representative of the 11 spans within the corridor. In each selected span, transects were established perpendicular to the length of the corridor, with transects starting in the treated area of the corridor, then traversing into the edge zone of the corridor (the zone where the treated areas meet the adjacent natural forest), and into the adjacent natural forest. These management zones are referred to as 'treated', 'edge', and 'forest' zones. The transects were 90 m in length, with 30 m in each management zone (Figure 1). In a few cases, the transects were extended to 94 m in length, where a vehicle track crossed the transect. Steel pickets were placed at the start and end of each transect, and these points were marked with a GPS, to allow the transects to be revisited over time.

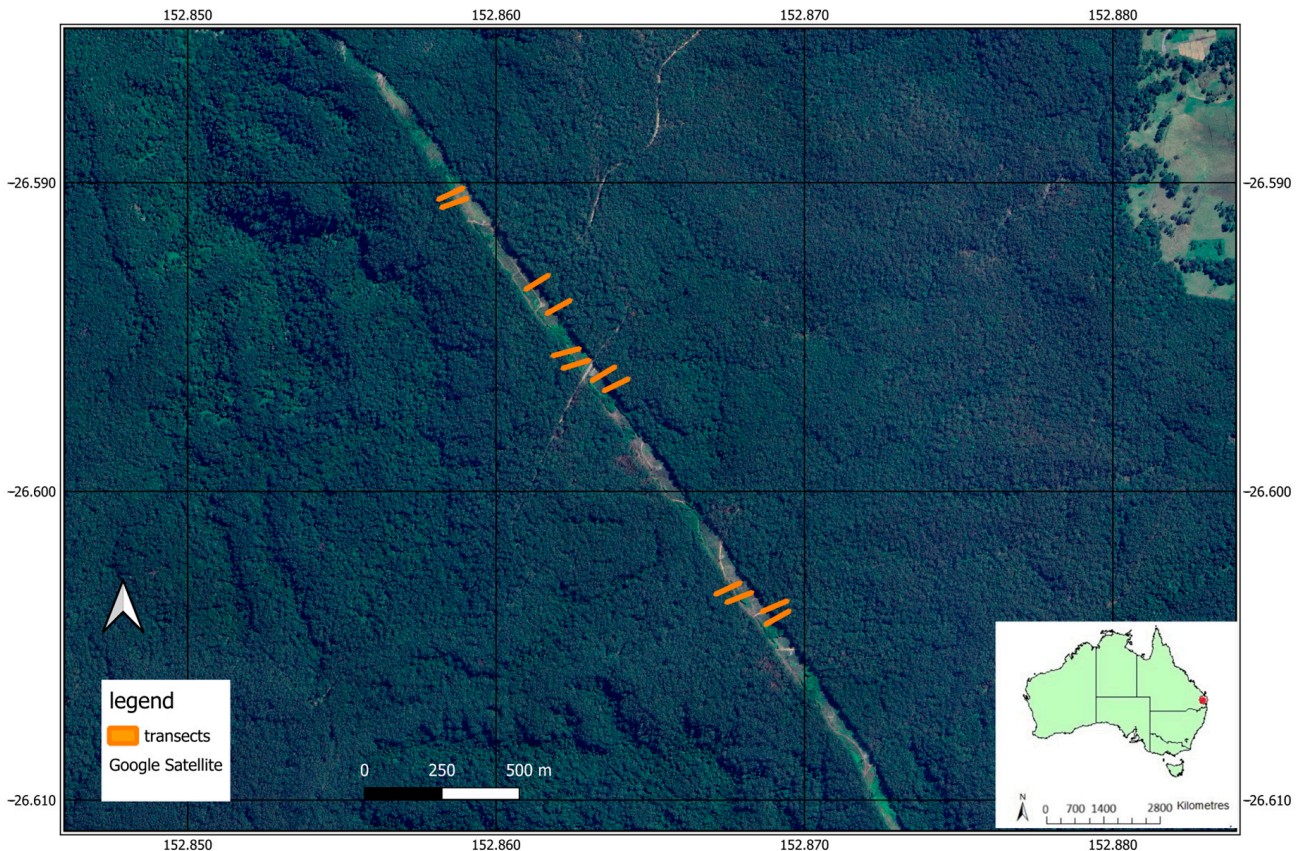

**Figure 1.** Locations of the 12 monitoring transects in the Mapleton Falls National Park transmission line corridor, Queensland, Australia.

In total, 12 transects were established to sample the variation in slope and aspect, with pairs of transects (mid-slope and upper slope) on alternating sides of the corridor (Figure 1). Effectively, there were 6 transects in each slope position (mid-slope and upper slope, as defined by Speight, 1990 [25]) and 6 transects in each orientation (north-east and south-west). Hence, there were three 'replicates' of each slope and aspect combination. Slope and aspect were included in this study as they can have an influence on vegetation assemblages [13,17,23]. Upper slope transects were located at least 20 m from tower structures to avoid the disturbed areas immediately around the towers. Sampling did not focus on the lower slope and gully positions, as these areas were generally not mega-mulched, as they are located in steep terrain and are less threat to the transmission line structures (as height from the canopy to the lowest catenary point is greatest in these areas). In some cases, transects crossed gullies that were adjacent to the corridor.

Data analysis focused on the differences in species composition and structure among the treated zones, edge zones, and natural forest areas. The influences of slope position and aspect were also investigated. A 15 × 2 m vegetation survey plot was located in each of the three management zones in each transect (Figure 2), where a full list of vascular plants was recorded. This plot size was deemed appropriate to sample most of the plant species within the immediate area. Species richness was calculated as the number of species occurring within this survey plot. These plots were located in the middle of each zone, at 7–22 m (treated zone), 37–52 m (edge zone), and 67–82 m (forest zone) on the central transect. Only plant species with their base within the survey plot areas were recorded. Plant species that could not be identified in the field were sent to the Queensland Herbarium for identification. Vegetation cover and height were recorded in four 1 × 1 m quadrats in each management zone (12 per transect, Figure 2). These quadrats were located at 6–7 m, 12–13 m, 18–19 m,

and 24–25 m in the treated zones, at 36–37 m, 42–43 m, 48–49 m, and 54–55 m in the edge zones, and at 66–67 m, 72–73 m, 78–79 m, and 84–85 m in the forest zones to adequately sample the variation along a given transect. Percentage cover was recorded for seven different groupings: (1) native perennial grasses, (2) native herbs, sedges, and ferns, (3) introduced species, (4) shrubs, (5) trees, (6) bare earth (and rocks), and (7) litter (debris). This was performed for different height classes, including (1) ground layer < 1 m height, (2) 1–3 m height, (3) 3–6 m height, and (4) tree layer > 6 m height. Heights were determined using four 1.5 m height sticks that could be joined to a total height of 6 m.

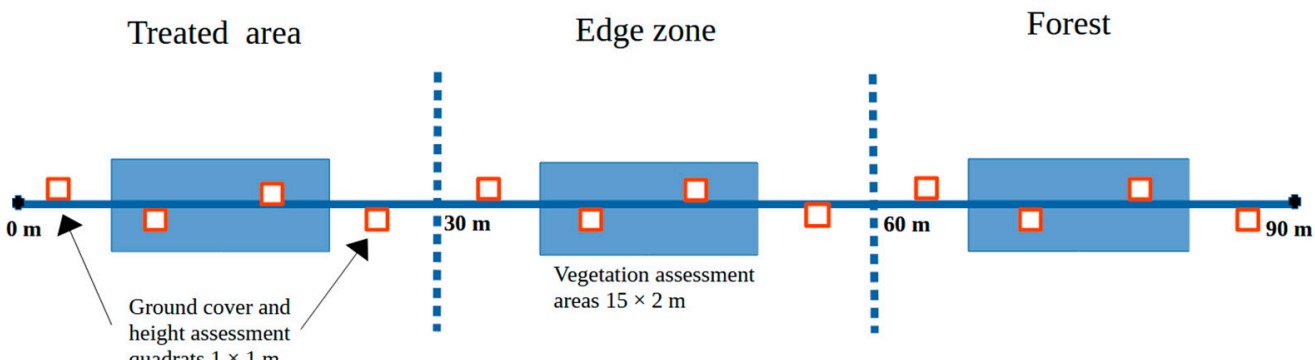

**Figure 2.** Layout of 90 m transects. Each transect traversed the treated (mega-mulched) area, edge zone, and forest area across the transmission line corridor. Three floristic survey areas (vegetation assessment areas) were located along the transect, with one in each management zone. Twelve 1 × 1 quadrats (four in each zone) were located along the transect to record vegetation cover at different heights.

To determine the influence of management zone, slope (mid-slope or upper slope), and aspect (transect orientation, north-east or south-west) on species richness and cover of different plant groups (native perennial grasses, native herbs, sedges and ferns, introduced species, shrubs, trees), bare earth, and litter, a two-factor analysis of variance was conducted using RStudio software (R version 4.3.1, 2023, The R Foundation for Statistical Computing). Tukey's Honestly Significant Difference post hoc tests were used to identify significant differences between management zones, where present. Regression relationships were also used to determine variation with distance along the transect (from middle of the treated area to the undisturbed forest). Variation in species composition was analysed using CANOCO Version 5. Plant presence/absence data were transformed using the logarithm ln (x + 1) where x = species presence, due to the presence of many zero values in the dataset. A constrained canonical variate analysis (CCA) was selected as the most appropriate form of multivariate analysis based on CANOCO adviser. Ordination biplots with species and environmental variables (e.g., management zones) were produced to visually show results from these analyses, with species represented by species codes outlined in the Supplementary Materials (Table S1). In addition to management zone, the influence of slope position, aspect, tree canopy cover, litter cover and depth, elevated vegetation cover (i.e., shrubs, tree regeneration), and near-surface vegetation cover (i.e., grasses, ferns, and low shrubs) on species composition was investigated.

At each transect in each zone (treated, edge, and forest), an overall fuel hazard rating was completed using the *Overall Fuel Hazard Assessment Guide* [26]. The mid-point in each management zone was used as the point at which to apply this rating. This rating is essentially a rapid visual assessment of tree bark fuels, elevated fuels (i.e., mid-layer vegetation), near-surface (i.e., ground-layer vegetation), and surface fuels (i.e., litter and debris) [26]. Where bark fuel refers to the different types of bark attached to tree trunks and branches, with varying levels of flammability. Elevated fuels are the shrubs, heath, and

suspended fuels that occur below the tree canopy, but above the near-surface fuels. The density and continuity (horizontal and vertical) of this layer has a strong influence on the spread of flames and flame heights. The near-surface fuels are those live and dead fuels just above the ground, that may make contact with the ground but are not lying on it. They include grasses, herbs, ferns, and low shrubs, and also have an influence of the rate of fire spread and flame heights. The surface fuels include leaves, twigs, bark, and other fine fuel lying on the ground, which influence the rate of spread of a fire. The guide [26] includes a series of tables and photographs to aid in the fuel hazard rating for each layer. Scoring incorporates the different fuel characteristics, such as cover, height, continuity, density, flammability (amount of dead material), and biomass. An overall score incorporating the different fuel components was calculated for each assessment point (as the sum of the influences of bark hazard, elevated fuel hazard, and the combined surface and near-surface hazard) so that comparisons of fuel hazard could be made between the treated and untreated vegetation. Scores of either low, moderate, high, very high, or extreme were made for each management zone of the transect, where these scores reflect likely variation in likely fire behaviour in the event of a wildfire and how easy it would be to control such a fire [26]. While this rating system has received some recent criticism in relation to its subjectivity and relationships to fire behaviour potential, it has been widely used by fire management agencies in Australia over the last decade [27].

A Scientific Research/Educational permit was submitted online through the Department of Environment and Science to allow for the collection of plant reference specimens. This permit was granted on 10 October 2023 (Authority number: P-PTUKI-100489824). Transect sampling took place between the 11 October and the 9 November 2023.

## 3. Results

### 3.1. Species Richness and Composition

A total of 96 different plant species were recorded across the 12 transects. A total of 65 species were recorded in the treated areas, 72 species were recorded in the edge zone, and 63 species were recorded in the forest zone. Average native plant species richness per 30 m$^2$ area was 20.3, 21.5, and 19.4 species in the treated, edge, and forest management zones, respectively (Figure 3). There was no significant difference in native species richness between the management zones (Figure 3, $F_{2,31} = 0.49$, $p > 0.05$). An analysis of variance also suggested there was no significant difference in species richness between different slope positions ($F_{1,31} = 0.31$, $p > 0.05$) or different aspects ($F_{1,31} = 1.08$, $p > 0.05$). Only six introduced species were recorded, and all of these occurred in very low abundance (often as a single individual in a survey plot). These species were recorded in the treated areas (*Passiflora suberosa*, *Emilia sonchifolia*, *Conyza bonariensis*, *Gomphocarpus physocarpus*, and *Lantana camara*) and in the forest (*Cinnamomum camphora*). A full list of the species recorded, with common names and functional group, is provided in Table S1 (Supplementary Materials).

The ten most frequently occurring species included *Entolasia stricta* (occurring at 94% of the sampling plots), *Pteridium esculentum* (occurring at 75% of the sampling plots), *Pultenaea villosa* (occurring at 72% of the sampling plots), *Austromyrtus glabra* (occurring at 69% of the sampling plots), *Lepidosperma laterale* (occurring at 69% of the sampling plots), *Acrotriche aggregata* (occurring at 69% of the sampling plots), *Callicoma serratifolia* (occurring at 64% of the sampling plots), *Elaeocarpus reticulatus* (occurring at 61% of the sampling plots), *Calochlaena dubia* (occurring at 58% of the sampling plots), and *Gahnia clarkei* (occurring at 58% of the sampling plots). Most of these species were particularly abundant in the treated zones, but often still occurred in the forest edge zone or forest zone at lower abundance. The canopy of the forest zones was dominated by *Eucalyptus pilularis* at all transects. Other

common canopy or sub-canopy species in the forest zone included *Eucalyptus racemosa*, *Syncarpia glomulifera*, and *Eucalyptus microcorys*. These tree species often also occurred in the edge zones, but with lower levels of canopy cover. *Callicoma serratifolia* was another common tree in the forest edge zones but was rarely greater than 10 m in height.

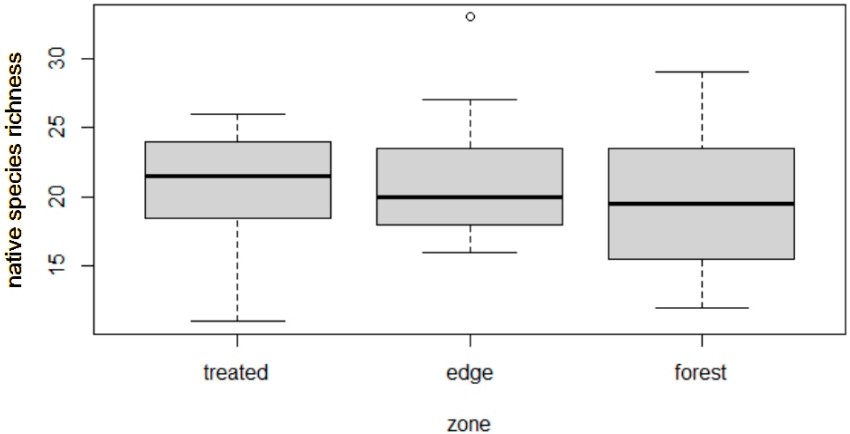

**Figure 3.** Boxplots of native plant species richness per vegetation survey plot (30 m$^2$) in the different management zones of the transects that traversed from the more intensively managed transmission line corridor (treated areas) to the less intensively managed edge zones and the adjacent natural forest.

After taking into account the correlations among environmental variables, three variables were found to have a significant influence on plant species composition. These were management zone (pseudo-F = 4.5, *p* = 0.002, explaining 11.7% of the variance in plant composition), litter depth (pseudo-F = 2.2, *p* = 0.002, explaining 5.6% of the variance in plant composition), and near-surface vegetation cover (pseudo-F = 2.0, *p* = 0.002, explaining 4.9% of the variance in plant composition). Slope position and aspect did not have a significant influence on plant composition in this study. Increasing litter depth was associated with the forest management zones, while increasing near-surface vegetation cover was associated with the treated management zones (Figure 4). There was a distinct group of plant species associated with the forest zones and higher levels of litter cover, including *Lophostemon confertus*, *Digitaria* sp., *Cissus hypoglauca*, *Eucalyptus microcorys*, *Archontophoenix cunninghamiana*, *Sticherus lobatus*, and *Endiandra discolor* (Figure 4). Plants associated with the treated zones with higher near-surface vegetation cover included *Dampiera sylvestris*, *Schoenus brevifolius*, *Oxalis chnoodes*, *Drosera spatulata*, *Gonocarpus tetrogynus*, *Dicranopteris linearis* var. *linearis*, *Gleichenia* sp., *Panicum simile*, *Pimelea linifolia*, *Logania albiflora*, *Pultenaea villosa*, *Acrotriche aggregata*, *Lepidosperma laterale*, *Pteridium esculentum*, and *Themeda triandra* (Figure 4). There was not a distinct group of species associated with the forest edge zone, as this zone tended to share species in common with the forest (e.g., *Eucalyptus pilularis*, *Dianella caerulea*, *Cassytha muelleri*, *Syncarpia glomulifera*) and the treated zone (e.g., *Astrotricha umbrosa*, *Goodenia rotundifolia*). Forest and edge zone plots tended to have greater similarity in species composition than plots in the treated management zone (Figure 4).

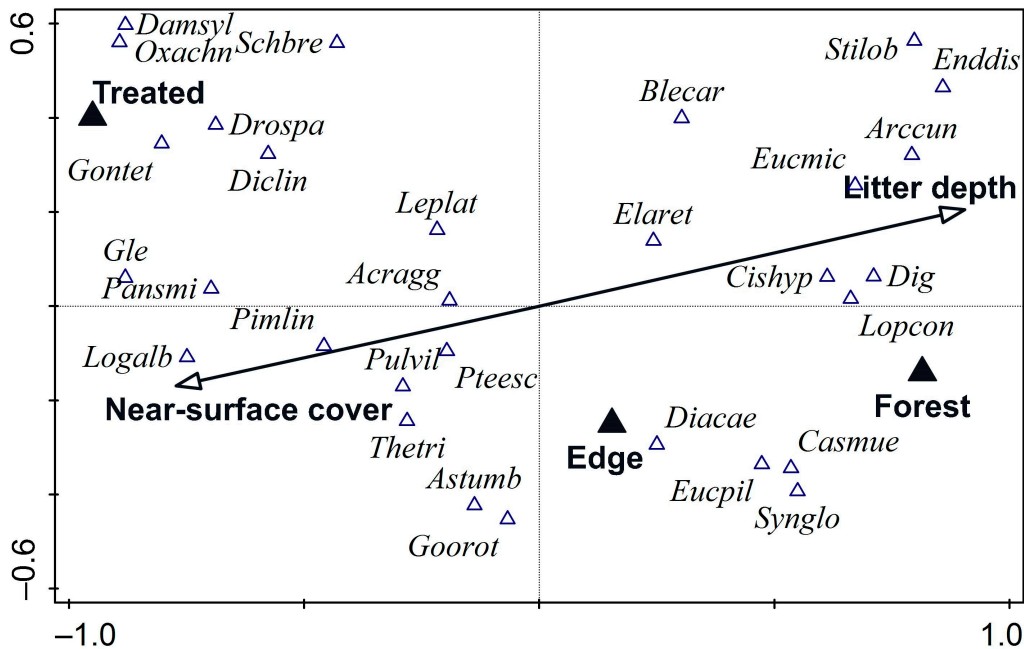

**Figure 4.** Ordination biplot showing species (unfilled triangles) associated with environmental variables that had a significant influence on plant composition (management zones, litter depth, and near-surface plant cover). Species names are abbreviated—refer to Table S1 for full species names and corresponding abbreviations. Arrows (representing environmental variables with numeric values) point in the direction of the steepest increase of environmental variable values. The angle between arrows indicates the correlation between individual environmental variables. Management zones (three different zones) are represented by filled triangles, where the distance between the symbols approximates the average dissimilarity in species composition among the sample classes (i.e., symbols that are closer are more similar in species composition).

### 3.2. Vegetation Cover and Structure

The cover of different plant groups < 1 m in height varied greatly among the management zones along a transect. Native grass cover differed significantly among management zones ($F_{2,31} = 42.90$, $p < 0.001$) but was not influenced by slope position or aspect ($p > 0.05$). Native grass cover was significantly higher in the treated zones relative to both the edge zones and the forest zones ($p < 0.001$, in both cases) but did not differ between the edge zone and the forest ($p > 0.05$, Figure 5). The cover of herbs, sedges, and ferns followed a similar trend to native grasses, differing significantly among the management zones ($F_{2,31} = 6.00$, $p = 0.006$), but with no influence of slope position or aspect ($p > 0.05$). The cover of herbs, sedges, and ferns was significantly higher in the treated zones relative to both the edge zones and the forest zones ($p = 0.044$ and $p = 0.006$, respectively) but did not differ between the edge zone and the forest ($p > 0.05$, Figure 5). The cover of introduced species was zero for all transects and hence was not analysed. The cover of shrubs was similar among the management zones ($p > 0.05$, Figure 5) and was not influenced by slope position or aspect ($p > 0.05$). However, shrubs 1–3 m in height tended to have higher levels of cover in the forest (2.4%) and the edge zone (1.6%), relative to the treated zone (0.7%).

The cover of bare earth and litter also varied among the management zones ($p = 0.012$ and $p < 0.001$, respectively) but were not significantly influenced by slope position or aspect ($p > 0.05$ in both cases). Bare earth cover was generally low across all management zones but was marginally higher in the treated zones than in the edge zones ($p = 0.068$), was significantly higher in the treated zones relative to the forest ($p = 0.012$), and did not differ between the edge zones and the forest (Figure 5). Litter cover was significantly lower in the treated zones, relative to the edge zone ($p < 0.001$) and the forest ($p < 0.001$), with the

edge zones having marginally lower litter cover than the adjacent natural forest ($p = 0.080$, Figure 5). Tree cover >6 m in height varied significantly among the management zones ($F_{2,31} = 154.9$, $p < 0.001$) but was not influenced by slope position or aspect ($p > 0.05$). Tree cover was zero in the treated areas and hence was significantly greater in both the edge zones and forest ($p < 0.001$, Figure 5). The forest also had a greater level of tree cover than the edge zone ($p < 0.001$, Figure 5).

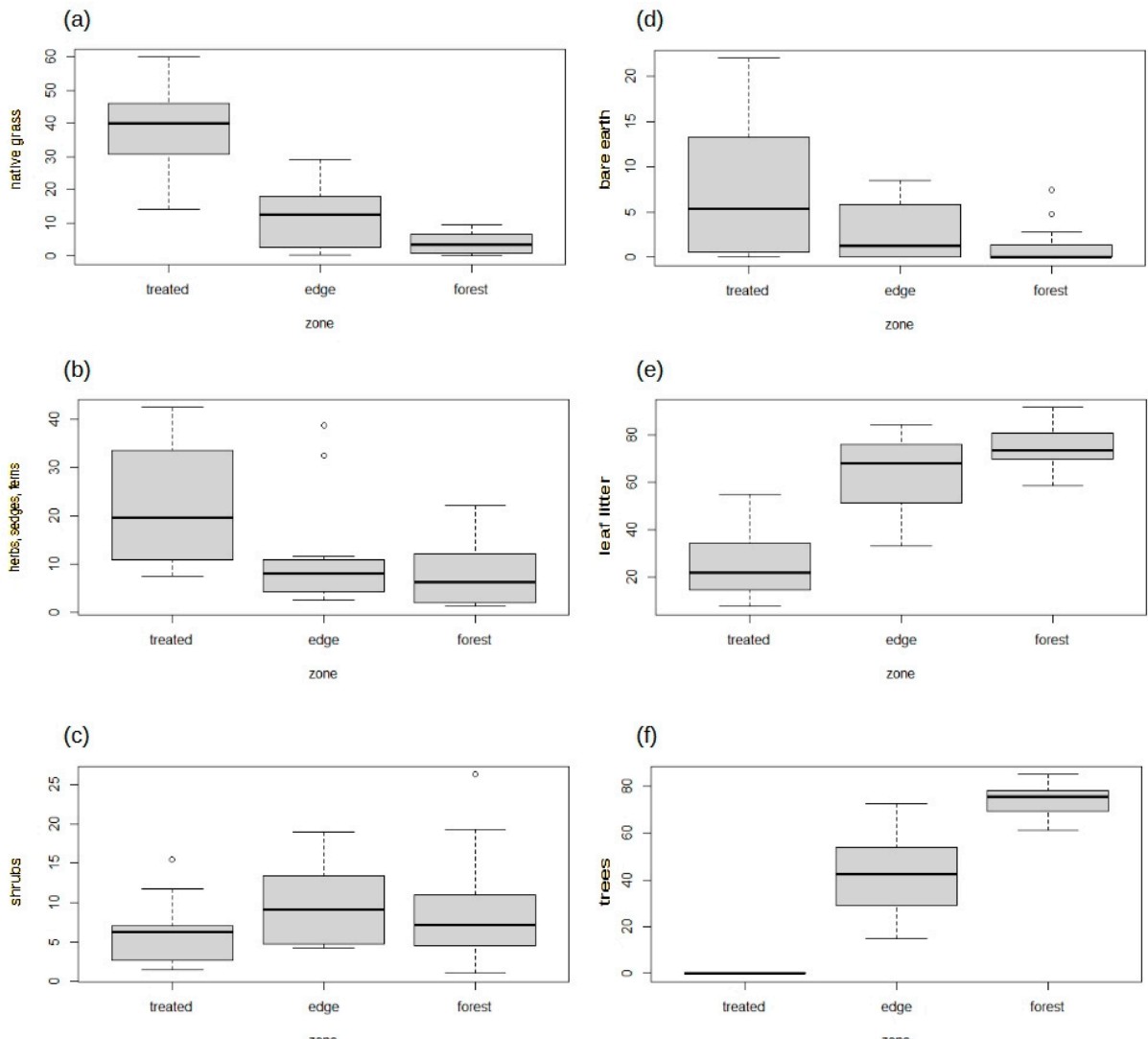

**Figure 5.** Boxplots of percentage cover of (**a**) native grasses, (**b**) herbs, sedges, and ferns, (**c**) shrubs, (**d**) bare earth, (**e**) leaf litter, and (**f**) trees, in the different management zones of the transects that traversed from the more intensively managed transmission line corridor (treated areas) to the less intensively managed edge zones and the adjacent natural forest.

Relationships between plant cover and distance along the transect were also analysed to investigate trends between the most heavily disturbed areas (beneath the transmission line infrastructure, start of transect) and the least disturbed areas (natural forest, end of transect). As expected, native grasses and native herbs, sedges, and ferns showed similar trends of declining cover with increasing distance along the transect (Figure 6). Tree cover (>6 m in height) and litter cover showed strong trends of increasing cover with increasing distance on the transect (Figure 7a,b). Shrub cover showed a weak positive trend of increasing cover along the transect (Figure 7c).

(a)

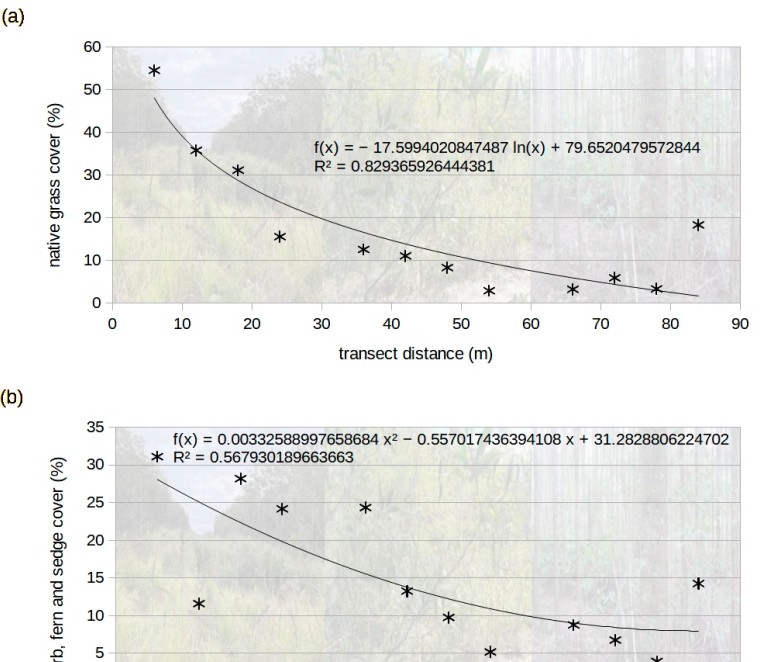

(b)

**Figure 6.** (**a**) Relationships between mean native grass cover and distance on the transect; (**b**) relationship between mean cover of native herbs, sedges, and ferns and distance on the transect. Where transects traversed from the more intensively managed transmission line corridor (0–30 m) to the less intensively managed edge zones (30–60 m) and the adjacent natural forest (60–90 m).

(a)

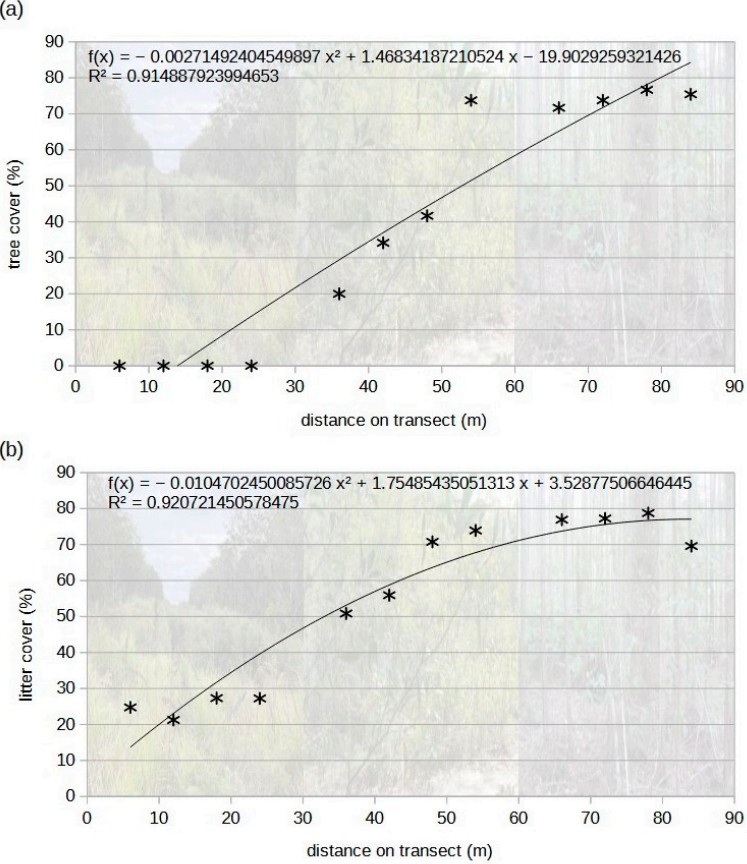

(b)

**Figure 7.** *Cont.*

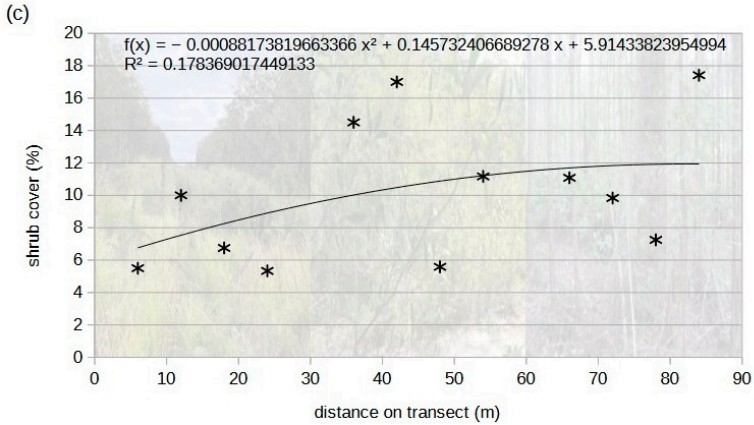

**Figure 7.** (**a**) Relationships between mean tree cover and distance on the transect; (**b**) relationship between mean litter cover and distance on the transect; (**c**) relationship between mean shrub cover and distance on the transect. Where transects traversed from the more intensively managed transmission line corridor (0–30 m) to the less intensively managed edge zones (30–60 m) and the adjacent natural forest (60–90 m).

### 3.3. Fuel Hazard Assessment

The overall fuel hazard varied greatly among the management zones. The fuel hazard was generally high to extreme across all transects, with no overall 'low' hazard values assigned. This was due to 'very high' or 'extreme' hazard levels in the combined surface and near-surface fuels across all management zones. However, the combined overall hazard was highest in the forest zones (with 75% of the transects assigned as 'extreme' hazard) and lowest in the treated zone (no 'extreme' hazard), with intermediate levels of hazard in the edge zones (25% of the transects with 'extreme' hazard) (Figure 8). The treated areas were assessed as typically having 'high' overall hazard (83% of the transects), while 'high' hazard was also common in the edge zones (50% of the transects), but less common in the forest areas (17% of the transects) (Figure 8). The reduction in overall hazard in the treated zones suggests that the various fuel management treatments (e.g., mega-mulching) have been successful in reducing fuel hazard, particularly through altering vegetation structure (e.g., removing bark fuels and the cover of elevated fuels).

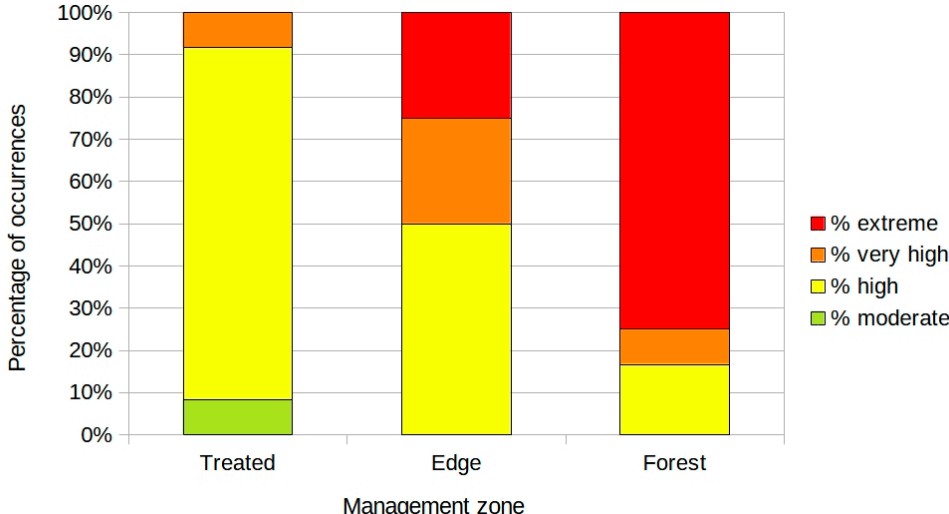

**Figure 8.** Overall fuel hazard (based on the *Overall Fuel Hazard Assessment Guide* [24]) in the different management zones of the transects. The overall hazard varied from 'moderate' to 'extreme' in the transects assessed and was generally lower in the treated management zones than the adjacent natural forest.

## 4. Discussion

Vegetation surveys showed clearly how the vegetation composition and structure has been modified by the management treatments in the transmission line corridor at MFNP. As hypothesized, the field measurements showed that there were clear differences in vegetation composition and structure between the treated corridor zones and the adjacent natural forest. The edge zones, where tall woody tree species had been manually felled or chemically treated but where management had been less intensive, also differed from both the treated zones and the adjacent forest, and had elements of both the treated zones and the adjacent forest. Hence, the management aim of modifying vegetation structure and composition in the treated zones to a state where herbaceous plant species were dominant, and where woody vegetation heights were reduced, has been achieved. This was demonstrated through significantly higher native grass cover and the cover of herbs, sedges, and ferns in the treated zones, and the lower cover of trees and tall woody plants.

Native herbaceous species that were most abundant in the treated zones, such as *Entolasia stricta*, *Pteridium esculentum*, *Lepidosperma laterale*, *Gahnia clarkei*, *Schoenus brevifolius*, and *Themeda triandra*, were typically still present in the adjacent natural forest at much lower abundance. This suggests that such species have proliferated in the disturbed areas in the absence of a tree canopy. There were also some species only recorded in the treated or edge zones of the corridor (e.g., *Gonocarpus tetrogynus*, *Dampiera sylvestris*, *Drosera spatulata*, *Panicum simile*, *Gleichenia* sp., *Dicranopteris linearis* var. *linearis*). This supports the literature that shows that transmission line corridors can provide novel habitats for early successional plant species [16,28]. The presence of a range of species that have benefited from the management disturbance in the transmission line corridor highlights the positive influence that such corridors can have on plant species diversity at a landscape scale. Similar findings are reported elsewhere; a range of light-demanding herbs, grasses, vines, and pioneer species are reportedly more abundant on the edges of corridors [13–15]. In fact, the ecological literature suggests that plant diversity is often higher on forest edges, because edges can contain a mix of plant species from the forest interior and the adjacent vegetation type [18–21]. This mixing of species from the treated zones and the natural forest was observed in the current study, although there were no significant differences in mean native species richness per survey area between the different management zones. Most studies suggest that edge effects on vegetation decline within 20 m of the forest edge [29–31]. This is in agreement with the findings of our study, where there were typically abrupt differences in vegetation cover variables between the treated zones and the forest zones, with the edge zones representing a transition between the two.

Despite the high level of disturbance associated with the mega-mulching treatments that have taken place in the transmission line corridor, the diversity of native species remained high, and there were very few introduced species occurring in the monitoring transects. Hence, our hypothesis that treatments would encourage introduced weed species was not supported. This finding is in contrast to other studies in Australia that have reported higher 'weediness' in transmission line corridors relative to the adjacent native vegetation [14,22] and at the forest edges more generally [21,23,32]. This positive outcome is likely a result of the position of the corridor within a National Park, with lower adjacent exotic plant seed banks. The high levels of native species cover (particularly grass cover) likely also helped reduce the opportunities for weed outbreaks at the site [33,34]. Nevertheless, there is still opportunity for the spread of weed seeds through the machinery and vehicles that visit the corridor. Vehicle hygiene and ongoing monitoring of weed species occurrences will be critical in ensuring that the corridor remains relatively weed free.

The changes in vegetation structure and composition in the transmission line corridor have resulted in a 'fire-break' through the National Park, where containment of a wildfire

might be possible under certain conditions, reducing the risk of a wildfire spreading through the National Park and to nearby private properties. The differences in vegetation structure among the management zones resulted in differences in overall fuel hazard ratings, with a lower overall fuel hazard in the treated zones relative to the edge zones and the natural forest. While it is recognised that the overall fuel hazard methodology is not an accurate measure of fuel quantity and fire behaviour potential [27], we believe the observed differences in fuel structure are indicative of reduced fuel hazard. This is encouraging for allowing future prescribed burning in the transmission line corridor. However, the overall fuel hazard in the treated zones was still assessed as 'high' for most transects (83%), due to the 'very high' or 'extreme' hazard levels in the combined surface and near-surface fuels. Lopes and Fernandes (2025) reported that, in resprouter-dominated communities, the recovery of fuel loads to approximately 10 t ha$^{-1}$ took place within three years after mechanical treatments [35], and a similar fuel load recovery is likely to be the case following treatments at MFNP [24], where most of the species can resprout vegetatively. Hence, prescribed burning should be conducted under mild weather conditions with adequate fuel moisture to ensure the fire can be easily managed. Operators should also be aware of the risk of creating an arcing pathway with smoke and ash ionizing the air gap between the vegetation and the transmission lines, which could lead to damage to the infrastructure or to human life [24,36].

Prescribed burning offers a relatively low-cost alternative vegetation treatment to replace mechanical treatments which are expensive to apply. A study of fuel dynamics in mechanically treated transmission corridors in Portugal [35] showed that fuel treatments were relatively short-lived, with a need to re-apply mechanical treatments at intervals of 3–10 years. Prescribed burns have been successfully applied along parts of the MFNP corridor, and further burns are planned for the future. However, while prescribed burning does offer a cost-effective solution to fuel management in the transmission line corridor, it is not without risk and, given the patchy nature of low-intensity fire, it will likely not kill all saplings. Hence, some chemical or mechanical follow-up management of woody vegetation is likely to be necessary. Livestock grazing (e.g., goats with virtual fencing) has been effectively used for managing fire prone vegetation in some cases (e.g., in Norway) [37], but such a management option is unlikely to be feasible in the National Park setting of the current study.

Despite the challenges of conducting prescribed fire in the transmission line corridor, the resulting influence of such fire is likely to be positive for the local native flora in the corridor. Numerous studies have demonstrated the benefits of regular prescribed burning to enhance the abundance and diversity of herbaceous vegetation [38–42]. There are also a number of relatively low-statured shrubs growing in the transmission line corridor that will benefit from fire, including obligate seeder species, such as *Pultenaea villosa*, *Pimelea linifolia*, and *Zieria minutiflora*.

Ongoing management of the transmission line corridor, regardless of its nature, will be critical in ensuring that the differences in vegetation structure and composition between the corridor and the surrounding forest are maintained over time. The presence of regenerating tree species, such as *Callicoma serratifolia*, even in the treated zones of the transects, indicates that such species will need to be managed once they grow to a certain height. Ongoing temporal monitoring or modelling of fuel dynamics [35] is needed to guide the maintenance schedule (e.g., timing of planned burns, the need for herbicide treatments, etc.) to help maintain low levels of hazardous fuels (i.e., biomass) whilst maintaining ground cover and biological diversity within the transmission line corridor. To efficiently determine changes in vegetation height over the corridor, without the need for transect monitoring, UAV (drone), or ground-based LiDAR (light detection and ranging), techniques could be applied

temporally (e.g., by extending the work by Sos et al., 2023) [24]. Further remote sensing work is also recommended to determine if spectral signatures can be used to identify key plant species (e.g., those considered a higher fire risk) occurring along the transmission line corridor [43,44].

## 5. Conclusions

In summary, management of the transmission line corridor at MFNP has resulted in pronounced differences in plant species composition and structure. The presence of a high diversity and cover of native species (including grasses, herbs, ferns, sedges, and shrubs) in the treated zones of the corridor and low abundance of introduced plant species is encouraging. Vegetation heights in the treated zone of the corridor rarely exceeded 2 m on the transects surveyed in this study but were greater than 30 m in the adjacent forest. Overall fuel hazard has been lowered as a result of the management practices. Ongoing monitoring is recommended to determine changes in plant species composition over time and changes in the vegetation heights over time, and to guide future fuel management treatments.

**Supplementary Materials:** The following supporting information can be downloaded at: https://www.mdpi.com/article/10.3390/fire8080309/s1, Table S1: Plant species recorded in monitoring transects, their abbreviations (species code) used in figures, common names, family, functional group, and number of occurrences.

**Author Contributions:** Conceptualization, T.L., S.M. and J.J.; methodology, T.L.; formal analysis, T.L.; investigation, T.L.; data curation, T.L.; writing—original draft preparation, T.L.; writing—review and editing, T.L., S.M. and J.J.; visualization, T.L.; project administration, T.L., S.M. and J.J.; funding acquisition, T.L., S.M. and J.J. All authors have read and agreed to the published version of the manuscript.

**Funding:** This research was funded by Powerlink Queensland. This research received no additional external funding.

**Institutional Review Board Statement:** Not applicable.

**Informed Consent Statement:** Not applicable.

**Data Availability Statement:** The data collected during this study is summarised in Table S1. Further inquiries can be directed to the corresponding author.

**Acknowledgments:** This project was funded by Powerlink Queensland. We are grateful for the assistance from Ruben Staal and Max for the day they spent assisting with fieldwork. The Department of Environment and Science provided permit approval for collecting herbarium samples for this project (AUTHORITY No. P-PTUKI-100489824).

**Conflicts of Interest:** Authors S.M. and J.J. are employed by Powerlink Queensland, the organisation which funded this study. T.L. is employed by Queensland Forest Consulting Services and he declares no conflicts of interest.

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
