# Peer review of "Modification of Vegetation Structure and Composition to Reduce Wildfire Risk on a High Voltage Transmission Line"

_fire, doi:10.3390/fire8080309_

Round 1

Reviewer 1 Report

Comments and Suggestions for Authors

Comments to: Modification of vegetation structure and composition to reduce fire risk on high voltage transmission line

The article presents a  post-treatment documentation of vegetation in a high voltage transmission corridor, focusing on vegetation composition and fire risk. Though there is little doubt that vegetation management, through reducing biomass, reduces fire risk, the balance between fire safety and ecological concerns requires attention from and cooperation between multiple stakeholders. Previous research in Australia and internationally indicates that non-native species are more common in the edges of man-made corridors compared to the interior of forests. The article aimed, among others, to clarify whether this was the case also in the studied corridor.  

The authors investigated how vegetation treatment of a 3 km long, 70 m wide transmission line corridor through Mapleton Falls National Park, Queensland, Australia, affected vegetation structure and composition. The aim of vegetation treatment in power line corridors is to keep vegetation low, so it shall not interfere with transmission cables, and reduce fire hazard from wildfires / bushfires. However, the creation of a low vegetation corridors in the tall eucalyptus forest may affect the composition of plants in the treated, edge and natural forest zones, for example because of more light, thus favoring some species. Previous research [20, 22] suggested that introduced species may take advantage of conditions generated by relatively open corridors. The authors therefore performed a field survey during late 2023, to identify present species, (and their abundance), three years after the massive treatment of the Mapleton Falls transmission line corridor in 2020.

Twelve transects were chosen, each 90 m long, perpendicular to the main direction of the transmission line corridor, consisting of 30 m “treated zone”, 30 m “edge zone” and 30 m “forest zone”. The transects were chosen to represent different slopes and aspects. The vegetation assessment areas, 15 x 2 m,  were located along the transects, with one assessment area in each management zone. Additionally, vegetation cover at different heights was studied in 12 quadrats, 1 x 1 m, 4 in each management zone.

A total of 96 different plant species were identified, in the 12 vegetation survey plots, of 30 m2 each. Of them, 65 were found in the treated zone, 72 in the edge zone and 63 in the forest zone. Only six introduced species were identified, in very low abundance. The fact that five of them were recorded in the treated areas, and one in the forest, may be in agreement with previous literature in the field [20, 22]. However, there is no knowledge of the vegetation composition before the treatments, so that it cannot be concluded with that the abundance of introduced weeds has increased. Differences in slope and aspect did not result in significant differences in species richness. The grass cover is highest in the treated zone, together with herbs, fern and sledge. Tree and litter cover is highest in the forest, while shrubs populate the area more evenly.

Suggestions for improvement

The subchapter 3.3 “Fuel hazard assessment”, should be extended to explain how the Overall Fuel Hazard Assessment Guide was used to produce the results on Figure 8. The respective paragraph in “Materials and Methods”, lines 198 – 208 is an introduction, but I believe that the readers of Fire will appreciate if you could exemplify the application of this methodology. An extension of this topic will create better balance between plant biology and fire safety focus, in the article.

Some expressions may be not understandable to all readers of Fire. Examples: What is an “open forest”? (lines 53 and 54). What is a “span”? (lines 126 – 128). Does “upper slope” translate “steep” and “lower slope” translate “nearly flat”?

In Norway is grazing with goats used (to some extent) to maintain low vegetation after treatment. A digital fencing technology keeps the animals in the corridor. The best time to introduce the animals is about three years after treatment, since the upcoming trees are very small, and easy for goats to consume. They prefer young trees compared to grass, so, they attack the trees targeted. The concern of introduced species could be addressed by placing the goats on a diet of only native foods, for example for one week before the grazing period starts. The treated area is 210 000,- m2. That would imply (for Norwegian conditions) that 210 goats have enough food for 1 month, or 70 goats for 3 months (more manageable).

Author Response

Reviewer 1.

Comments to: Modification of vegetation structure and composition to reduce fire risk on high voltage transmission line

The article presents a post-treatment documentation of vegetation in a high voltage transmission corridor, focusing on vegetation composition and fire risk. Though there is little doubt that vegetation management, through reducing biomass, reduces fire risk, the balance between fire safety and ecological concerns requires attention from and cooperation between multiple stakeholders. Previous research in Australia and internationally indicates that non-native species are more common in the edges of man-made corridors compared to the interior of forests. The article aimed, among others, to clarify whether this was the case also in the studied corridor.

The authors investigated how vegetation treatment of a 3 km long, 70 m wide transmission line corridor through Mapleton Falls National Park, Queensland, Australia, affected vegetation structure and composition. The aim of vegetation treatment in power line corridors is to keep vegetation low, so it shall not interfere with transmission cables, and reduce fire hazard from wildfires / bushfires. However, the creation of a low vegetation corridors in the tall eucalyptus forest may affect the composition of plants in the treated, edge and natural forest zones, for example because of more light, thus favoring some species. Previous research [20, 22] suggested that introduced species may take advantage of conditions generated by relatively open corridors. The authors therefore performed a field survey during late 2023, to identify present species, (and their abundance), three years after the massive treatment of the Mapleton Falls transmission line corridor in 2020.

Twelve transects were chosen, each 90 m long, perpendicular to the main direction of the transmission line corridor, consisting of 30 m “treated zone”, 30 m “edge zone” and 30 m “forest zone”. The transects were chosen to represent different slopes and aspects. The vegetation assessment areas, 15 x 2 m, were located along the transects, with one assessment area in each management zone. Additionally, vegetation cover at different heights was studied in 12 quadrats, 1 x 1 m, 4 in each management zone.

A total of 96 different plant species were identified, in the 12 vegetation survey plots, of 30 m2 each. Of them, 65 were found in the treated zone, 72 in the edge zone and 63 in the forest zone. Only six introduced species were identified, in very low abundance. The fact that five of them were recorded in the treated areas, and one in the forest, may be in agreement with previous literature in the field [20, 22]. However, there is no knowledge of the vegetation composition before the treatments, so that it cannot be concluded with that the abundance of introduced weeds has increased. Differences in slope and aspect did not result in significant differences in species richness. The grass cover is highest in the treated zone, together with herbs, fern and sledge. Tree and litter cover is highest in the forest, while shrubs populate the area more evenly.

Response 1: Thanks for your review and positive feedback.

Suggestions for improvement

The subchapter 3.3 “Fuel hazard assessment”, should be extended to explain how the Overall Fuel Hazard Assessment Guide was used to produce the results on Figure 8. The respective paragraph in “Materials and Methods”, lines 198 – 208 is an introduction, but I believe that the readers of Fire will appreciate if you could exemplify the application of this methodology. An extension of this topic will create better balance between plant biology and fire safety focus, in the article.

Response 2: This is a fair point. We have added some more details on the fuel hazard methodology and its widespread use in Australia. Refer to page 8, lines 223-247.

Some expressions may be not understandable to all readers of Fire. Examples: What is an “open forest”? (lines 53 and 54). What is a “span”? (lines 126 – 128). Does “upper slope” translate “steep” and “lower slope” translate “nearly flat”?

Response 3: We have accordingly revised these expressions, and provided references to support their use, where necessary. Wording changed on: page 2, line 58; page 5, line 138; and page 6, line 159. A new reference has been added regarding slope position. These positions don’t translate to differences in steepness. They relate to positioning in a topo-sequence, relative to crests, flats and depressions. Effectively the lower-slope position is closest to a depression or gully, the upper-slope position is closest to the crest and the mid-slope position is in between.

In Norway is grazing with goats used (to some extent) to maintain low vegetation after treatment. A digital fencing technology keeps the animals in the corridor. The best time to introduce the animals is about three years after treatment, since the upcoming trees are very small, and easy for goats to consume. They prefer young trees compared to grass, so, they attack the trees targeted. The concern of introduced species could be addressed by placing the goats on a diet of only native foods, for example for one week before the grazing period starts. The treated area is 210 000,- m2. That would imply (for Norwegian conditions) that 210 goats have enough food for 1 month, or 70 goats for 3 months (more manageable).

Response 4: This is interesting to hear about. We have taken this more as a general comment rather than a comment that needs to be addressed in the revision. However, we have now included a reference to the use of livestock in managing fuel loads in the discussion (page 19, lines 470-472).

Reviewer 2 Report

Comments and Suggestions for Authors

This is an interesting study about the assessment of fuel treatments to mitigate wildfire risk. In general, the manuscript is well written and structured, but it has some shortcomings as follows:

Abstract: The abstract has several shortcomings in terms of methodology and presentation of key results.

Materials and Methods: Briefly describe the method of assessing the fuel risk.

In the Discussion: Compare the main results of this research with other related previous research.

Additional comments are presented in the text.

Given these shortcomings, the manuscript requires minor revisions.

Author Response

Reviewer 2.

This is an interesting study about the assessment of fuel treatments to mitigate wildfire risk. In general, the manuscript is well written and structured, but it has some shortcomings as follows:

Abstract: The abstract has several shortcomings in terms of methodology and presentation of key results.

Response 1: We appreciate the positive feedback. We believe that the shortcomings in the abstract and results have now been addressed in the comments below.

Materials and Methods: Briefly describe the method of assessing the fuel risk.

Response 2: We have added additional information to provide a summary of this method in the revised manuscript (page 8, lines 223-247).

In the Discussion: Compare the main results of this research with other related previous research.

Response 3: We believe that we had, for the most part, adequately compared our results with previous research throughout the discussion. However, we have added some further comparisons to research done on this topic (e.g. page 17, lines 410-411; page 18, lines 452-456; page 19, lines 462-465 and 470-473).

Additional comments are presented in the text.

Given these shortcomings, the manuscript requires minor revisions.

Page 1, Line 3 ‘use wildfire instead’

Response 4: Agree. Change has been made, as suggested (page 1, line 3).

Page 1, Line 15-17. ‘modify the sentence to clarify better the issue’

Response 5: We have made some changes to this sentence (page 1, lines 15-17) to improve clarity.

Page 1, Line 17-18. ‘After defining the objective, the study method is not mentioned in the abstract.’

Response 6: We believe our methodology has been described in the abstract: page 1, lines 18-23: “We measured vegetation structure (cover and height) and composition (species presence in 15 × 2 m plots) at 12, 90 m transects on the transmission line corridor, to determine if management goals were being achieved and to determine how the vegetation and fire hazard (based on the Overall Fuel Hazard Assessment method) varied among the treated corridor, the forest edge environment and the natural forest”. We have added some additional wording to this sentence, but were hesitant to add too many words due to word limit restrictions of the abstract.

Page 1, Line 21-23. ‘The result is very general written. Express the results quantitatively and qualitatively’

Response 7: Some additional information has been added to the abstract results in the revised manuscript (page 1, lines 29-31, page 2, lines 34-35).

Page 1, Line 27. ‘It is not clear how to assess the risk of fire or fuels’

Response 8: In the revised manuscript we have now mentioned in the abstract that this was based on the Overall Fuel Hazard Assessment method (page 1, lines 21-22), and have provided some more in-depth results on this (page 2, lines 34-35).

Page 2, Line 45. ‘You can use its abbreviation in the following’

Response 9: Agree, with have abbreviated this in the revised manuscript (pages 2, 3, 4, 5, 17, 18, 19, 20).

Page 3, Line 64. ‘better use fuel treatment’

Response 10: Agree, with have modified the wording in this case (page 3, line 70).

Page 4, Line 115. ‘Uniformity should be observed in the text. Various combinations have been used in this paper, such as vegetation management treatments’

Response 11: While ‘fuel’ and ‘vegetation’ management treatments are essentially the same thing in this context, we have tried to be consistent and use ‘fuel management’ where possible, throughout the manuscript (pages 3, 4, 19, 20).

Page 4, Line 121. ‘A brief history of historical fires in this region should be mentioned’

Response 12: Some information on the fire history of the National Park has now been added on page 4, lines 130-136.

Page 7, Line 198-199. ‘It is better to provide a brief explanation of this hazard assessment method here’

Response 13: This is a fair point. While it is difficult to summarise a 42 page guide, some additional information has been added to give the reader a better understanding of the methodology, without having to refer to the original report. Changes have been made on page 8, lines 223-247.

Page 8, Line 216. ‘It is not clear in the methodology which index was used to determine species richness.’

Response 14: Agree, this was an oversight. We have added some wording to clarify this metric. Refer to page 6, lines 174-175.

Page 12, Lines 297-298 Figure 5. ‘The titles of the axes in the graphs, especially the vertical axis, are somewhat unclear’

Response 15: Agree. We have changed the vertical axis labels now to match those in the figure caption and improve clarity (page 12, Figure 5).

Page 16, Lines 353-384. ‘Instead of a general explanation of the research findings, which are mentioned in detail in the results section, here, while mentioning the key results, an analytical comparison is made with the findings of others’

Response 16: Somewhat agree. We believe that this first paragraph of the discussion can be used as a summary of the results in relation to the aims and hypotheses of the study. Hence we have opted not to make an ‘analytical comparison’ to other studies in this first paragraph. In the following paragraph we have added some additional comparison to other studies (page 17, lines 410-411), although we note that we had already referred to other studies in this paragraph (page 17, lines 416, 418, 422). We have also made reference to additional studies at other points in the discussion section (e.g. page 18, lines 452-456; page 19, lines 462-465 and 470-473).

Reviewer 3 Report

Comments and Suggestions for Authors

Make the explanation clearly and easily understood 

Comments on the Quality of English Language

After the article revised, it could be published

Author Response

Reviewer 3.

This case study was located in Mapleton Falls National Park, south-east Queensland, Australia

>>>Is there any reasons why the research located in Mapleton Falls National Park?

Response 1: This area was selected for our case study, as it had received recent and well documented fuel management treatments: “An adaptive management trial, as a collaboration between Powerlink Queensland and the Queensland Parks and Wildlife Service was initiated in 2018….” (page 3, lines 75-79). Additionally it had been used for an earlier study, mentioned in the introduction section (i.e. Sos et al. (2023), page 4, line 109). Additional wording has been added: page 4, lines 129-130.

The transmission line corridor through Mapleton Falls National Park is approximately 70 m

wide and runs approximately 3 km in length.

>>> Is there any reasons why the corridor has 70 m wide and runs approximately 3 km in

length ?

Response 2: This is simply based on the length of the corridor that traverses the National Park, and operational requirements regarding corridor widths. Because of the height of the adjacent forest (i.e. approx. 30-35 m) it is important that trees that fall into the corridor (e.g. during storms) do not make contact with the infrastructure. This is not something that we had any control over in the research design and is descriptive information only.

This study focused on a selection of six tower spans.

>>>Why only six towers ?

Response 3: We focused on six spans, as this was feasible with the budget allocated for this project. We believe this provides an adequate representation of the tower spans in the National Park (of which there are 11 in total). Additional wording has been added, page 5, lines 142-143.

In total 12 transects were established to sample the variation in slope and aspect, with pairs of

transects (mid-slope and upper-slope) on alternating sides of the corridor (Figure 1)

>>>Why only 12 transects were established to sample the variation in slope and aspects ?

Response 4: This was simply what was deemed feasible with the budget allocated for this project and was considered adequate to cover the variation in vegetation, slope and aspect of the site. We don’t feel that it is necessary to justify this within the manuscript.

Effectively, there were six transects in each slope position (mid-slope and upper-slope) and six

transects in each orientation (north-east and south-west). Hence there were three ‘replicates’ of

each slope and aspect combination. Upper-slope transects were located at least 20 m from tower

structures, to avoid the disturbed areas immediately around the towers.

>>> Why only six transects in each slope position (mid-slope and upper-slope) and six transects

in each orientation (north-east and south-west) ?

Response 5: Again, this was simply what was deemed feasible with the budget allocated for this project and was considered adequate to cover the variation in vegetation, slope and aspect of the site.

>>> Why only three replicates ?

Response 6: Three ‘replicates’ was considered the minimum number for us to be able to test for statistical differences among management zones, slope positions and aspects. As with most research projects there were time and budgetary constraints, which limited the number replicates in this study. In the ideal world more replicates would have been added, and we would include replication across different corridors in the region. However, we are confident in the results presented, and observations suggest that the findings are representative of this corridor.

>>>> Why Upper-slope transects were located at least 20 m from tower structures ? is there

any clear reason ?

Response 7: Yes, and we believe this is clearly stated on page 6, line 163: “to avoid the disturbed areas immediately around the towers.”

Data analysis focused on the differences in species composition and structure among the treated

zones, edge zones and natural forest areas. The influences of slope position and aspect were

also investigated.

>>>>Why focused on the differences in species composition and structure among the treated

zones, edge zones and natural forest areas ?

Response 8: We believe the introduction provides the necessary background on why we focused on differences among these different zones (pages 3-4, lines 95-106). The literature suggests that treated zones can have a higher abundance of non-native weed species relative to adjacent forest, as per our hypothesis (page 4, lines 124-126). Edge zones were also of interest as they represent the inter-face between the treated zones and the natural forest.

>>>> Why the influences of slope position and aspect were also investigated ? Any clear

explanation ?

Response 9: Yes. Ecological literature shows that both slope and aspect can have an influence on vegetation assemblages. For example, the Beer (1999) study and Elgegard et al. (2015) study found aspect had an influence on forest edge composition. We have added some additional wording to the manuscript to provide this justification for including these variables in the study (page 6, lines 161-162).

A 15 × 2 m vegetation survey plot was located in each of the three management zones in each

transect (Figure 2), where a full list of vascular plants was recorded.

>>>>Why used a 15 x 12 m vegetation survey plot ?

Response 10: We opted to use a 15 m long survey plot to fit within the middle of the 30 m zone of each transect. This plot size was deemed appropriate to sample most of the species in the area, with the time restrictions for sampling the vegetation. Numerous studies have evaluated plot size and no consistent recommendations are made in the literature on this. Typically, 1 × 1 m quadrats are used for herbaceous vegetation communities. We have added some additional wording on page 6, lines 173-174.

These plots were located at 7–22 m (treated zone), 37–52 m (edge zone) and 67–82 m (forest

zone) on the central transect. Only plant species with their base within the survey plot areas

were recorded. Plant species that could not be identified in the field were sent to the Queensland

Herbarium for identification.

>>>> These plots were located at 7–22 m (treated zone), 37–52 m (edge zone) and 67–82 m

(forest zone) on the central transect ?

Response 11: Yes, this was to ensure each plot was located in the middle of each zone. Additional wording has now been added to manuscript to clarify this (page 6, line 176).

>>>>Why Only plant species with their base within the survey plot areas were recorded ?

Response 12: This is common practice for plot-based vegetation surveys. We don’t feel that it is necessary to justify this within the manuscript.

Vegetation cover and height were recorded in four 1 × 1 m quadrats in each management zone

(12 per transect, Figure 2). These quadrats were located at 6–7 m, 12–13 m, 18–19 m, and 24–

25 m in the treated zones, at 36–37 m, 42–43 m, 48–49 m and 54–55 m in the edge zones, and

at 66–67 m, 72–73 m, 78–79 m, 84–85 m in the forest zones.

>>>> Why using 1m x 1 m quadrat in each management ?

Response 13: This quadrat size is typically used for recording vegetation cover, particularly in herbaceous ecosystems. The BioCondition Assessment Manual (Eyre et al. 2015) uses quadrats of this size for sampling ground-cover attributes across a range of ecosystems in our region. Reference: Eyre, T. J., Kelly, A. L., Neldner, V. J., Wilson, B. A., Ferguson, D. J., Laidlaw, M. J., & Franks, A. J. (2015). BioCondition: A condition assessment framework for terrestrial biodiversity in Queensland. Assessment Manual. Version 2.2. Information Technology, Innovation and the Arts, 81. We don’t feel that it is necessary to justify this within the manuscript.

>>>>Why those quadrats located and given fixed distance ?

Response 14: We opted to locate the quadrats at fixed distances, to adequately sample the variation along a transect (wording added on page 6, lines 183-184). This allowed investigation of regression relationships to determine variation with distance along the transect (i.e. from middle of treated area to undisturbed forest) (page 7, line 205-206).

Percentage cover of native perennial grasses, native herbs, sedges and ferns, introduced

species, shrubs, trees, bare earth (and rocks) and litter (debris) was recorded. This was done in

different height classes: (1) ground layer 6 m height. Heights were determined using four 1.5

m height sticks, that could be joined to a total height of 6 m. Photographs were taken at the 15

m, 45 m and 75 m points on each transect.

>>>> What use 8 different classes ? Any clear explanation ?

Response 15: Actually, seven different groupings were used here. The wording of this sentence has been modified to improve the clarity (page 6, line 184-186). These groupings (or very similar ones) are commonly used in vegetation surveys.

>>>> For calculating the height why only 1.5 m stick ?

Response 16: Multiple 1.5 m height sticks are used for ease of transport between sites. As stated in the manuscript (page 7, lines 188-189): “Heights were determined using four 1.5 m height sticks, that could be joined to a total height of 6 m.”

>>>>Is there any reasons why photograph were taken 15 m, 45 m and 75 m on each transect

Response 17: Taking photographs is common practice during vegetation surveys. However, as no photographs were used in the manuscript we have decided to delete this sentence (page 7, line 190).